# Development of Photothermal Catalyst from Biomass Ash (Bagasse) for Hydrogen Production via Dry Reforming of Methane (DRM): An Experimental Study

**DOI:** 10.3390/molecules28124578

**Published:** 2023-06-06

**Authors:** Ittichai Kanchanakul, Thongchai Rohitatisha Srinophakun, Sanchai Kuboon, Hiroaki Kaneko, Wasawat Kraithong, Masahiro Miyauchi, Akira Yamaguchi

**Affiliations:** 1Interdisciplinary of Sustainable Energy and Resources Engineering, Faculty of Engineering, Kasetsart University, Bangkok 10900, Thailand; ittichai.ka@ku.th; 2Department of Chemical Engineering, Kasetsart University, Bangkok 10900, Thailand; 3National Nanotechnology Center National Science and Technology Development Agency (NSTDA), Pathum Thani 12120, Thailand; sanchai@nanotec.or.th (S.K.); wasawat@nanotec.or.th (W.K.); 4Department of Materials Science and Engineering, Tokyo Institute of Technology, Tokyo 152-8550, Japan; hkaneko@ceram.titech.ac.jp (H.K.); mmiyauchi@ceram.titech.ac.jp (M.M.); ayamaguchi@ceram.titech.ac.jp (A.Y.)

**Keywords:** hydrogen production, photothermal catalysis, dry reforming of methane, biomass waste, bagasse ash

## Abstract

Conventional hydrogen production, as an alternative energy resource, has relied on fossil fuels to produce hydrogen, releasing CO_2_ into the atmosphere. Hydrogen production via the dry forming of methane (DRM) process is a lucrative solution to utilize greenhouse gases, such as carbon dioxide and methane, by using them as raw materials in the DRM process. However, there are a few DRM processing issues, with one being the need to operate at a high temperature to gain high conversion of hydrogen, which is energy intensive. In this study, bagasse ash, which contains a high percentage of silicon dioxide, was designed and modified for catalytic support. Modification of silicon dioxide from bagasse ash was utilized as a waste material, and the performance of bagasse ash-derived catalysts interacting with light irradiation and reducing the amount of energy used in the DRM process was explored. The results showed that the performance of 3%Ni/SiO_2_ bagasse ash WI was higher than that of 3%Ni/SiO_2_ commercial SiO_2_ in terms of the hydrogen product yield, with hydrogen generation initiated in the reaction at 300 °C. Using the same synthesis method, the current results suggested that bagasse ash-derived catalysts had better performance than commercial SiO_2_-derived catalysts when exposed to an Hg-Xe lamp. This indicated that silicon dioxide from bagasse ash as a catalyst support could help improve the hydrogen yield while lowering the temperature in the DRM reaction, resulting in less energy consumption in hydrogen production.

## 1. Introduction

Substitutional energy sources have drawn the public’s attention in recent years due to the consequences of fossil fuel consumption. One of the alternative energy sources is hydrogen (H_2_), an energy carrier that allows energy transport in a usable form from one place to another [1]. Clean hydrogen can be produced in several ways, such as through electrolysis. However, one of the lucrative approaches is the dry reforming of methane (DRM) process, which uses greenhouse gases as feedstock to produce synthesis gases (H_2_ and CO). There are a few drawbacks to the DRM reaction. First, it needs more than 700 °C to reach optimal efficiency and conversion [2]. This is energy intense. Second, the coking formation occurs at high temperatures. Several transition metals and noble metals, including Rh, Ni, Ir, Ru, Pt, and Co, are known to be active as DRM catalysts. The active metals are usually dispersed as small (nanoscale) particles supported on porous ceramic supports, such as Al_2_O_3_, and SiO_2_ [3], through catalytic synthesis methods. However, Ni is the most suitable active metal due to its comparatively lower cost to upscale production to the industrial scale. Despite the economical price of Ni, Ni catalysts suffer inevitably from rapid deactivation caused by coke deposition, active metal sintering, or both [4], and it is less stable compared to the other noble catalysts. The rate of carbon deposition was reported to decrease with rising reaction temperature [5]. However, the increment in temperature is not a reasonable way to stop carbon deposition due to energy use concerns. Several researchers reported an improvement of the temperature reduction in thermal catalysts for the DRM reaction, such as Rh, Ni, Pd, Co, Ir, and Ce supported on Al_2_O_3_ and SiO_2_ [6,7,8,9]; nonetheless, these require a trade-off with the higher cost of noble metals. Recently, a unique way of designing catalysts using plasmonic nanoparticles (NPs) has appeared to be an attractive approach for the DRM reaction to reduce the operating temperature. The plasmonic/metal NPs interact with light incidents, such as sunlight or light sources with heat, by transferring photoexcited charge carriers from metal NPs to the reactants, leading to chemical transformations under less energy-intense conditions. In this case, it is ideally possible to target electronic excitation so that only DRM reactions are activated. This leads to sustainability goals by lowering the operating temperature that traditionally runs at high temperatures and by improving the selectivity of reactions that may undergo side reactions.

Sugar is one of the top export products from Thailand. However, despite all the profits from the sugar industry, sugar production generates massive waste materials, such as bagasse, press mud, and spent wash [10]. Biomass waste from sugar production, such as bagasse, could be used as fuel for thermal power plants or boiler stations to produce energy that can be fed back into sugar production. However, residuals ash from burning bagasse would be disposed of at landfills, which could bring about other environmental issues caused by the bagasse ash.

The current study focused on utilizing waste materials (bagasse ash) and modifying their properties as a catalyst support for photothermal catalyst in the DRM reaction compared with a commercial support catalyst (SiO_2_). Furthermore, we used a conventional wet impregnation approach to obtain Ni/SiO_2_ catalysts in a synthesis design strategy.

## 2. Results and Discussion

### 2.1. Fabrication of the Catalyst Support Preparation and Synthesis Catalyst

In this work, catalyst support was prepared by extracting SiO_2_ from bagasse ash using an acidic extraction approach (3% HCl reagent grade), then modifying its properties by KOH activation at a ratio of 1:4 [11] to maximize the surface area of extracted SiO_2_. For catalyst synthesis, Ni/SiO_2_ was synthesized by conventional wet impregnation as shown in Figure 1. There are two mechanisms of wet impregnation. One relies on capillary action to draw the solution into the pores. The other is that the solution transport changes from a capillary action process to a diffusion process in the wet impregnation method [12].

### 2.2. Characterization

The synthesized catalyst surface area was analyzed using an Autosorb^®^ iQ3 gas sorption analyzer (Anton Paar QuantaTec Inc.; Boynton Beach, FL, USA), in which adsorption-desorption isotherms take place with liquid nitrogen’s help at −195 °C. The Brunauer-Emmett-Teller (BET) technique was used to calculate the catalyst surface area within the pressure range of 0.12 to 0.20. X-ray powder diffraction (XRD) patterns were recorded using a diffractor (XRD; Rigaku, Smartlab; Tokyo, Japan) axial diffractometer in the 2θ = 10° to 80° angular arrays with a step of 0.05°s^−1^ and CuK-α1 radiation (the wavelength CuK-α1 = 1.5406 Å) of the diffractometer for the synthesized catalysts. Then, the synthesized catalysts were analyzed using X-ray fluorescence (Epsilon 1; Malvern Panalytical Ltd.; Malvern, UK) for the elemental analysis to confirm the chemical composition of the catalysts in percentage terms. The UV–vis DRS were measured using a JASCO V-670 spectrometer; JASCO Coorperation; Tokyo, Japan in the wavelength range from 200 to 800 nm with BaSO_4_ as the reference. Additionally, scanning electron microscopy (SEM) images of the synthesized catalysts were measured using Schottky field emission scanning electron microscopy (FE-SEM; SU8030; Hitachi-High Tech Corp.; Tokyo, Japan). The SEM images captured the surface morphology of the catalysts, and the EDS mapping checked the dispersion of Ni particles and the composition of the catalysts. Transmission electron microscopy (TEM) images were taken on a Jeol-JEM-2100Plus; JEOL Ltd.; Tokyo, Japan operated at 200 kV. Specimens were prepared by suspending sample powders in ethanol; then, a drop of the suspension was deposited on copper grids.

#### 2.2.1. XRD Analysis

As shown in Figure 2a, extracted SiO_2_ was extracted by acidic extraction using HCl and followed by KOH activation at various ratios 1:2–1:6. These samples were characterized by X-ray diffraction analysis. The amorphous SiO_2_ can be detected at 24.3°. The ratio at 1:4 exhibited a high surface area and average pore size [11], which could be the optimal ratio of SiO_2_: KOH for KOH activation. Additionally, the XRD pattern of 3Ni/SiO_2_ BA WI shows in Figure 2b, obtained by the Wet Impregnation approach. For a fresh 3Ni/SiO_2_ catalyst (light green line), the diffraction peaks appearing at 37.1°, 43.1°, and 62.5° can be attributed to the NiO phase (JCPDS 65-2901), while the broad peak at 24.3° can be identified to the SiO_2_ phase (JCPDS 39-1425) [13]. The XRD pattern of the reduced 3Ni/SiO_2_ catalyst (blue line) also shows in Figure 2b which the diffraction peaks at 2θ = 44.3°, 51.4°, and 76.1°, which can be indicated by the crystal planes of (111), (200), and (220) of metallic nickel phase. After the catalytic activity test, the reduced 3Ni/SiO_2_ WI catalyst after the DRM process (red line) shows that the XRD pattern demonstrated identically to the reduced 3Ni/SiO_2_ WI catalyst (blue line). Therefore, we suspected coke formation after the DRM reaction. However, the reduced 3Ni/SiO_2_ WI catalyst after the DRM process (red line) exhibits XRD pattern without the appearance of the diffraction peaks of carbon.

#### 2.2.2. BET Surface Analysis

The BET analysis results are shown in Table 1. The BET surface area of the commercial SiO_2_ was 10.7 m^2^/g, which changed slightly to 11.4 and 12.6 m^2^/g in the 3 and 5 Ni/SiO_2_ commercial WI samples, respectively. The BET surface area of the extracted SiO_2_ with KOH activation at a ratio 1:4 was 185 m^2^/g, which marginally declined to 163 and 157 m2/g in the 3 and 5 Ni/SiO_2_ BA WI, respectively (Table 1). From the Barrett–Joyner–Halenda (BJH) method, the average pore sizes of the commercial SiO_2_, the extracted SiO_2_ with KOH activation, 3 and 5 Ni/SiO_2_ commercial WI, and 3 and 5 Ni/SiO_2_ BA WI were 4.47, 20.2, 4.62, and 20.9 nm, respectively. The pore sizes of samples can play a crucial role in the diffusion of CH_4_ and CO_2_ molecules. In fact, pore size influences the diffusion of reactant molecules to the catalytically active sites within the catalyst material. If the pore size is too small, it can restrict the movement of reactants, leading to slower reaction rates.

#### 2.2.3. SEM Analysis

As shown in Figure 3, the internal microstructures of the bare commercial SiO_2_, extracted SiO_2_, and the Ni/SiO_2_ commercial WI were examined using SEM analysis, while EDS elemental mapping was used for the Ni/SiO_2_ BA WI. The bare commercial SiO_2_ is shown in Figure 3a, with a regular, smooth surface. Figure 3b shows an image of the extracted SiO_2_ from BA with a complicated structure and a rough surface. Figure 3c,d represent synthesized Ni/SiO_2_ commercial WI and Ni/SiO_2_ BA WI, respectively. As seen, the Ni particles were successfully decorated on both the commercial SiO_2_ and extracted SiO_2_ BA surfaces. Furthermore, the extracted SiO_2_ BA exhibited a rough structure with a high surface area, which would be beneficial for photothermal catalytic activity. The EDS element spectra of the Ni/SiO_2_ BA composite are shown in Figure 3e, which confirmed not only the presence of Ni, Si, and O but also residual inorganic elements, such as Mg, Al, Fe, and K, on the synthesized catalyst. Figure 3f demonstrates the dispersion of Ni particles onto the catalyst support (SiO_2_).

#### 2.2.4. XRF Analysis

Despite the EDS elemental mapping, XRF analysis helped to estimate the composition of the synthesized catalyst. The XRF analysis results indicated that Ni and SiO_2_ were not only in the obtained samples, but there were also other residual elements, such as Al and K. In contrast, the XRF analysis of Ni/SiO_2_ commercial WI only exhibited Ni and Si, as shown in Table 2. However, XRF analysis results did not include oxygen atoms in its calculation; thus, recalculating with the oxygen atoms will result in a 3% nickel by weight percentage.

#### 2.2.5. Optical Properties

The optical properties of the synthesized photothermal catalysts were evaluated using ultraviolet-visible spectroscopy. The UV–Visible diffuse reflectance spectrum (UV–Vis DRS) of bare SiO_2_ commercial, extracted SiO_2_ BA, 3 and 5 Ni/SiO_2_ commercial WI, and 3 and 5 Ni/SiO_2_ BA WI are shown in Figure 4a,b. The UV–Vis DRS of the commercial SiO_2_ had a non-absorption edge at all wavelengths (blue DRS line in Figure 4a). In contrast, after the wet impregnation method, 3 and 5 Ni/SiO_2_ commercial WI exhibited an increment of absorption edge around 370 nm and strong absorption over a wide range of the UV and visible light regions (black and red DRS lines in Figure 4a). At the same time, the UV–Vis DRS of the extracted SiO_2_ from BA had an absorption edge around 230 nm (blue line in Figure 4b). It is because not only extracted SiO_2_ from bagasse ash contain pure SiO_2_, but it also consists of inorganic residuals such as Al, K, Mg, and Fe, which could add light adsorption properties to our extracted SiO_2_ sample. Furthermore, after the wet impregnation method, 3 and 5 Ni/SiO_2_ BA WI demonstrated a similar trend to 3 and 5 Ni/SiO_2_ commercial WI, with an increment of strong absorption over a wide range of the UV and visible light regions (black and red DRS lines in Figure 4b).

### 2.3. Photothermal Catalytic Activity

#### 2.3.1. Photothermal Catalytic Hydrogen Generation Results and Analysis

The photothermal catalytic activities of the synthesized catalysts were examined based on CH_4_, CO_2_ conversion, and H_2_ yield under UV–visible light irradiation. We investigate the efficient catalytic activity between different catalyst supports (commercial SiO_2_ and extracted SiO_2_ from bagasse ash); their catalytic performance toward H_2_ generation was carried out under UV–visible light from a Hg-Xe lamp. The light-adsorption ability of commercial SiO_2_ demonstrates in Figure 4a blue line indicating that commercial SiO_2_ cannot interact with UV–visible light. However, the light-adsorption ability is improved after the wet impregnation method, as shown in Figure 4a (red and black line). At the same time, the light-adsorption ability of the extracted SiO_2_ BA was also enhanced by the wet impregnation method, which resulted in light-adsorption ability from 200 nm to 800 nm (Figure 4b red and black line) compared to the bare extracted SiO_2_ (Figure 4b blue line) before the wet impregnation method, which only absorbed light in a range of 200–350 nm. Therefore, the catalysts could interact with UV–visible light after the wet impregnation method. Figure 5 shows the relative intensity of the Hg-Xe lamp, which was intense in the 200–450 nm range. The biomass-derived catalysts (Ni/SiO_2_ BA WI) had higher activity than the commercial SiO_2_ catalyst support one, as shown in Figure 6a,b because the surface area of Ni/SiO_2_ BA WI was substantially higher than for the Ni/SiO_2_ commercial WI, which could be attributed to higher activity.

#### 2.3.2. Band Alignment and Proposed Photothermal Catalytic Mechanism

Amorphous silica has a wide band gap energy of approximately 7.62–9.70 eV [14]; thus, the valence band electrons are relatively difficult to excite to the conduction band even when irradiated by UV light. However, the interband excitation in Ni particles was more favorable, with the photogenerated hot electrons overcoming the energy barrier. The previous study suggests that the hot carriers generated from the light-induced d-to-s interband excitation in Ni nanoparticles are proposed to mediate the transformation of photon energy to chemical energy [15,16].

To investigate the optimal Ni percentage amount on the synthesized catalysts with light irradiation, the photothermal catalytic performance of 3 and 5 Ni/SiO_2_ BA WI and 3 and 5 Ni/SiO_2_ commercial WI were tested in relation to the CH_4_ and CO_2_ conversions. Figure 6a,b revealed that 3Ni/SiO_2_ BA WI had the highest catalytic activity. However, when the content of Ni increased to 5%, the CO_2_ conversion and H_2_ yield were reduced, perhaps due to the agglomeration of nickel particles on the catalyst support. In addition, the light adsorption property of Ni/SiO_2_ BA WI could have increased the surface temperature on the catalyst support, leading to sintering, which subsequently resulted in catalyst deactivation and reduced 5Ni/SiO_2_ BA WI performance.

In contrast, the CH_4_ conversion of all synthesized catalysts showed a downward trend, indicating that the CH_4_ reactant was consumed in the system, as shown in Figure 6d. An increase in the temperature resulted in reduced CH_4_ conversion. Furthermore, H_2_O appeared in the system.

The experimental results demonstrated that Ni particles on extracted SiO_2_ from bagasse ash could interact with UV light to generate hydrogen at 300 °C because the UV–visible light provides photon energy to stimulate the Ni particles, leading to a plasmonic effect in the metal nanoparticles that creates electron-hole pairs (called hot carriers) and initiates the DRM reaction (Equation (1)). However, when the temperature increased, the H_2_ yield dropped to 450 °C, and H_2_O was detected in the system. Preferable reaction pathways, such as a reverse water gas shift reaction (Equation (2)) at low temperatures (200–350 °C), may have resulted in a side reaction and unwanted products, such as H_2_O in this case.
CH_4_ + CO_2_ ↔ 2H_2_ + 2CO_2_ ΔH°_298K_ = +247 kJ mol^−1^(1)
CO_2_ + H_2_ ↔ CO + H_2_O ΔH°_298K_ = +41 kJ mol^−1^(2)

## 3. Materials and Methods

### 3.1. Preparation of Bagasse Fly Ash

Bagasse fly ash was received from A Sugar Production Company (Thailand) after being burned as biomass fuel for the boiler. It contained a high moisture percentage; thus, it was necessary to dry the bagasse fly ash at 80 °C in a dry oven overnight, followed by drying at 105 °C in an oven for 2 h. After drying, the bagasse fly ash was fed into a crucible and burnt in a furnace at an initial temperature of 400 °C, a heating rate of 1 °C/min, and a holding time of 2 h to de-volatized any organic compounds. Then, the ash was heated to 900 °C to de-carbonize it and held for 2 h at the same heating rate to obtain crystalline silica.

### 3.2. Preparation of Silicon Dioxide Extraction

Bagasse fly ash was washed with 3% HCl reagent grade at a ratio of 12 mL 3% HCl per 1 g of bagasse fly ash as an extraction agent to reduce impurities other than SiO_2_ in the bagasse fly ash. The washed ash was stirred using a magnetic stirrer at 240 rpm for 2 h on a hotplate at 200 °C. After mixing, the sample was cleaned using deionized water until the pH was neutral. Afterward, the sample was passed through a vacuum filter containing filter papers and dried at 80 °C in a dry oven overnight. Then, the sample was calcined at 400 °C at a heating rate of 1 °C/min and cooled in the oven.

### 3.3. Preparation of KOH Activation for Silicon Dioxide

Silica dioxide is chemically activated by potassium hydroxide (KOH) at SiO_2_-to-KOH ratios ranging from 1:1 to 1:8 to obtain SiO_2_ particles with greater surface areas. The resulting mixed SiO_2_ samples (~3 g) and KOH (1:1 to 1:8) were added to 100 mL of DI water and then stirred at 70 °C for 1 h. After mixing, the samples were dried in the oven at 80 °C overnight until all the liquid had evaporated. Subsequently, each sample was placed in a ceramic crucible and calcined under an air atmosphere at 800 °C (heating rate 5 °C/min) for 1 h. During the calcination process, porosity was induced in the silica structure by the combustion of K_2_O derived from the KOH reaction. Next, the activated products were stirred with 2.5% HCl reagent grade to eliminate residual K_2_SiO_3_ in the samples. Afterward, the mixture was washed with DI water repeatedly to dissolve any KCl until the pH was neutral, followed by vacuum filtration to separate the solids from the liquid. Finally, the solid product was dried at 80 °C overnight to obtain the activated SiO_2_ with a high surface area.

### 3.4. Preparation of Catalysts

#### Wet Impregnation Method

Nickel(II) nitrate hexahydrate (Ni(NO_3_)_2_·6H_2_O) is dissolved with DI water to obtain a Nickle nitrate solution. Then, a Nickle nitrate solution is dropped on the activated SiO_2_ powder and mixed with a spatula. Subsequently, the samples are dried at 100 °C overnight and calcined at 600 °C for 2 h to obtain a Ni/SiO_2_-WI. Before use, a fresh catalyst was reduced with H_2_ reduction treatment at 600 °C for 2 h.

### 3.5. Photothermal DRM Activity Test

The photothermal activities of the powder samples were measured under ambient pressure in a flow reactor with a quartz window [17], which enabled us to irradiate the powder samples with a 150 W Hg–Xe lamp (Hayashi-Repic, LA-410UV-5; Tokyo, Japan). Approximately 10 mg of catalyst powder was put into a reactor; in sequence, the gas mixture CH_4_:CO_2_:Ar = 1:1:98 in vol% was continuously supplied to the reactor at a flow rate of 10 mLmin^−1^. The generated hydrogen was measured using a micro gas chromatograph (Agilent, 3000 Micro GC; Santa Clara, CA, USA).

## 4. Conclusions

The Ni particles on extracted SiO_2_ from bagasse ash (Ni/SiO_2_ BA WI) and commercial SiO_2_, fabricated using acidic extraction and KOH activation, could drive the photothermal catalytic DRM reaction under UV light. Compared to Ni particles on commercial SiO_2_ (Ni/SiO_2_ commercial WI) with light irradiation, the Ni/SiO_2_ BA WI could generate more H_2_ yield. In addition, light irradiation lowered the initiation temperature of syngas generation to 300 °C. However, the maximum H_2_ yield was only 3%. As a proposed assumption, the hot carriers generated from light-induced d to s interband excitation in the Ni particles played a vital role in our study, mediating the transformation of photon energy to chemical energy and driving the DRM reaction, despite the side reactions, such as the reverse water gas shift reaction. Therefore, we suggest that the use of extracted SiO_2_ from bagasse ash as the catalyst support provides a perspective on substitute materials for a practical path to establishing a plasmonic or non-plasmonic hot carrier-based photothermal catalytic system. The concept presented in this study showed the possibility of the utilization of waste materials as a solution for lowering reaction temperature and utilizing UV–visible light to activate the dry reforming of methane, which leads to energy reduction in its process. We expect that this concept will contribute to the progress of green energy and photothermal catalysis in the field of heterogeneous catalysis.

## Figures and Tables

**Figure 1 molecules-28-04578-f001:**
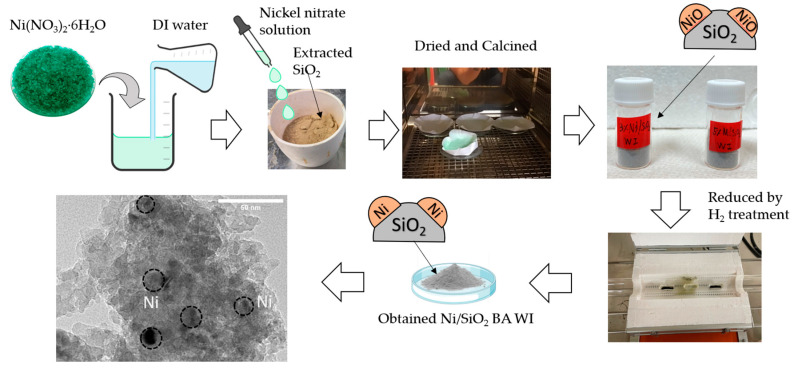
Schematic diagram of Ni/SiO_2_ bagasse ash (BA) catalysts prepared using wet impregnation method (WI).

**Figure 2 molecules-28-04578-f002:**
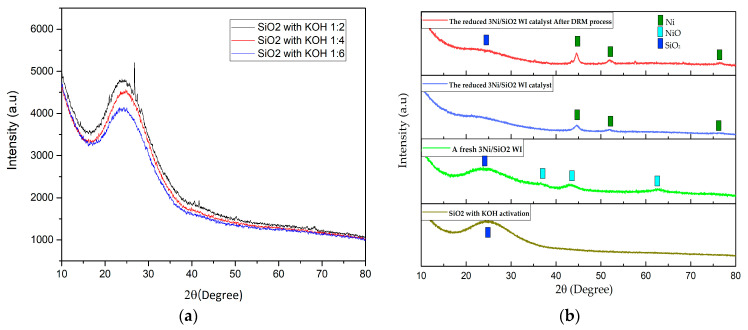
(**a**) The XRD pattern of extracted SiO_2_ with KOH activation at various ratios of SiO_2_ and KOH, (**b**) The XRD pattern from top to bottom of the reduced 3Ni/SiO_2_ after the DRM process, the reduced 3Ni/SiO_2_ WI fresh 3Ni/SiO_2_ BA WI, a fresh reduced 3Ni/SiO_2_ WI and SiO_2_ after KOH activation.

**Figure 3 molecules-28-04578-f003:**
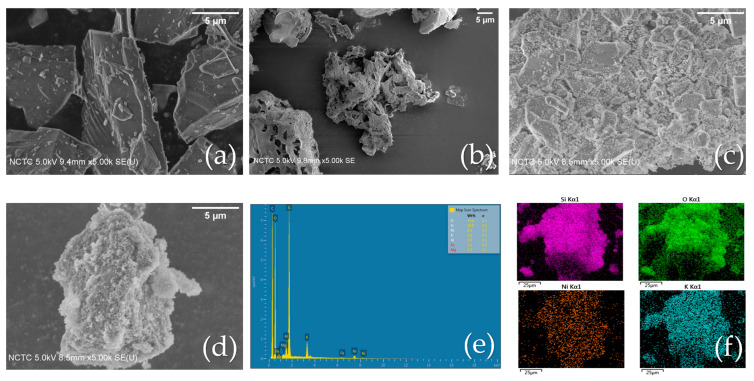
SEM images of (**a**) bare commercial SiO_2_, (**b**) extracted SiO_2_ from bagasse ash, (**c**) Ni/SiO_2_ commercial WI, (**d**) Ni/SiO_2_ BA WI. (**e**) EDS analysis of Ni/SiO_2_ BA WI. (**f**) EDS mapping of Ni/SiO_2_ BA WI.

**Figure 4 molecules-28-04578-f004:**
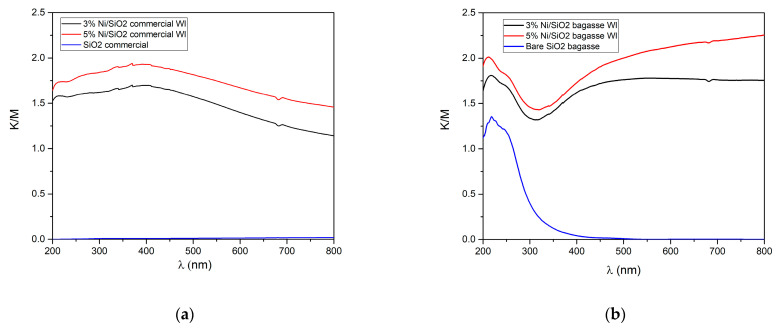
(**a**) UV–vis spectra of commercial SiO_2_, 3 and 5 Ni/SiO_2_ comm WI, (**b**) UV–vis spectra of extracted SiO_2_ from bagasse ash, 3 and 5 Ni/SiO_2_ BA WI.

**Figure 5 molecules-28-04578-f005:**
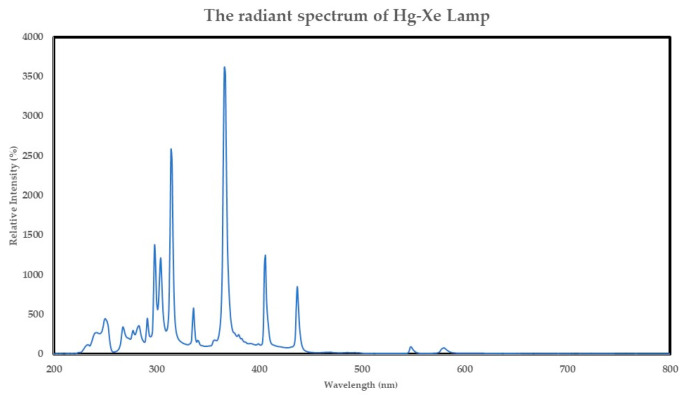
The radiant spectrum of Hg- Xe lamp.

**Figure 6 molecules-28-04578-f006:**
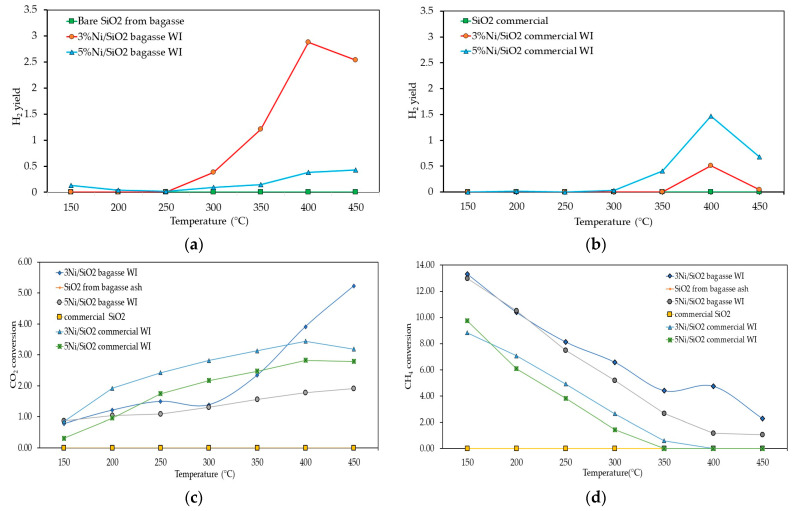
Photothermal catalytic activities of the synthesized catalysts at 4 h reaction time (**a**) H_2_ yield of Ni/SiO_2_ from bagasse ash (**b**) H_2_ yield of Ni/SiO_2_ commercial (**c**) CO_2_ conversion of synthesized catalysts (**d**) CH_4_ conversion of synthesized catalysts.

**Table 1 molecules-28-04578-t001:** Physicochemical properties of extracted SiO_2_ and Ni/SiO_2_ WI catalyst.

Samples	BET ^1^ Surface Area (m^2^/g)	Average Pore Size (nm)
Bare commercial SiO_2_	10.7	4.47
Bare extracted SiO_2_ from bagasse ash	42.3	36.0
Extracted SiO_2_ BA, KOH activation 1:2	207	13.1
Extracted SiO_2_ BA, KOH activation 1:4 *	185	20.2
Extracted SiO_2_ BA, KOH activation 1:6	178	11.6
3Ni/SiO_2_ commercial WI	11.4	4.62
5Ni/SiO_2_ commercial WI	12.6	4.85
3Ni/SiO_2_ bagasse ash WI	163	20.9
5Ni/SiO_2_ bagasse ash WI	157	20.9

^1^ The Brunauer-Emmett-Teller (BET) technique. * This ratio is used as catalyst support for catalysts synthesis.

**Table 2 molecules-28-04578-t002:** The elemental composition of 3Ni/SiO_2_ BA WI and 3Ni/SiO_2_ commercial WI.

Sample	Compounds			
	Si	Ni	Al	K
3Ni/SiO_2_ BA WI	71.4%	11.7%	5.26%	11.7%
3Ni/SiO_2_ commercial WI	88.0%	11.7%	-	-

## Data Availability

The data presented in this study are available on request from the corresponding author. The data is not publicly available due to the program agreement between the author (student) and the university.

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
