# Peer review of "Development of Photothermal Catalyst from Biomass Ash (Bagasse) for Hydrogen Production via Dry Reforming of Methane (DRM): An Experimental Study"

_molecules, 2023, doi:10.3390/molecules28124578_

Round 1
Reviewer 1 Report
Please see the attachment

Author Response
Dear Reviewer 1,
I would like to send you the review answer.
if you have further questions, please ask.
please find the attached file.
Best regards,
ITTICHAI KANCHANAKUL

Reviewer 2 Report
In this manuscript nickel based catalysts supported on silica from bagasse ash are studied for photothermal dryforming of methane with an aim of low-temperature reaction. The obtained catalyst samples are characterized using XRD, N2 adsorption, SEM, XRF, UV-Vis spectra and photothermal catalytic activity evaluation. 3%Ni/SiO2 BA shows higher hydrogen yield at 300 °C.
Several questions are as follows.
1 Table 1, extracted SiO2 shows high surface area than commercial SiO2. But commercial SiO2 can possess as surface area as several hundred m2/g. Why commercial SiO2 with 10.7 m2/g is used.
2 Table 2, the concentration of Ni of Ni/SiO2 BA WI is 11.7%, for which sample, which is far higher than that of either 3%Ni/SiO2 BA or 5%Ni/SiO2 BA.
3 Figure 6, what are the reaction times for the data. And Figure 6(d), why does the CH4 conversion decrease as the temperature rises, which is thermally catalyzed or photocatalyzed or photothermally catalyzed? A controlled experiment at heating but without light irradiation is needed.
4 Lines 276-281, are the catalyst samples reduced before reaction? If not, how do they thermally catalyze the reaction.
The expression is acceptable.
Author Response
Dear Reviewer 2,
I would like to send you the answer for your comment.
if ou have further questions, please ask again
please find the attached file.
best regards,
ITTICHAI KANCHANAKUL

Round 2
Reviewer 1 Report
Please find the attachent

Author Response
Dear Reviewer 1,
I would like to send you the answer of second round review.
please find the attached flie.
Best regards,
ITTICHAI KANCHANAKUL

Reviewer 2 Report
The initial manuscript was reviewed on May 2. Four suggestions were given for reference, which are not explained in this manuscript. In addition, Figure 1 provided is not mentioned in the body. So, the experiumental data obtained by the authors are not corresponding well to the title.
Author Response
Dear Reviewer 2,
I would like to send you the answer for your comment.
please find the attaced file.
BEst regards,
ITTICHAI KANCHANAKUL
